# Microbial, Physicochemical Profile and Sensory Perception of Dry-Aged Beef Quality: A Preliminary Portuguese Contribution to the Validation of the Dry Aging Process

**DOI:** 10.3390/foods12244514

**Published:** 2023-12-18

**Authors:** Ana Ribeiro, Irene Oliveira, Kamila Soares, Filipe Silva, Paula Teixeira, Cristina Saraiva

**Affiliations:** 1CECAV–Veterinary and Animal Science Research Centre, University of Trás-os-Montes e Alto Douro (UTAD), 5000-801 Vila Real, Portugal; anajacinta83@gmail.com (A.R.); kamilas@utad.pt (K.S.); fsilva@utad.pt (F.S.); 2AL4AnimalS—Associate Laboratory for Animal and Veterinary Sciences, 5000-801 Vila Real, Portugal; 3Department of Veterinary Sciences, School of Agrarian and Veterinary Sciences, University of Trás-os-Montes e Alto Douro (UTAD), 5000-801 Vila Real, Portugal; 4Department of Mathematics, School of Science and Technology, University of Trás-os-Montes e Alto Douro (UTAD), 5000-801 Vila Real, Portugal; ioliveir@utad.pt; 5EMAT (Center for Computational and Stochastic Mathematics), IST-UL, 1049-001 Lisboa, Portugal; 6CBQFCentro de Biotecnologia e Química Fina—Laboratório Associado, Escola Superior de Biotecnologia, Universidade Católica Portuguesa, Rua Diogo Botelho 1327, 4169-005 Porto, Portugal; pcteixeira@ucp.pt

**Keywords:** dry aged beef, aging process, microbiological profile, sensory analysis

## Abstract

Beef dry-aging consists of a selection of unpackaged prime cuts placed in a controlled environment cold room for several weeks. The goals are to concentrate flavors like nutty and beefy and to improve tenderness. The aim of this study was to verify the microbiological and physicochemical behavior and sensory perception of meat during a sample process example of meat dry-aging. Twelve beef loins were selected for 90 days of dry aging and placed in a cold room with average temperature, relative humidity and forced air 3.2 ± 0.7 °C, 60.7 ± 4.2% and 0.5–2 m/s, respectively. Samples of crust and lean meat were collected on days 1, 14, 21, 35, 60 and 90 of the dry aging process for microbiological, physicochemical (pH, a_w_, color L*a*b*) and sensory analysis. During drying, no pathogenic bacteria were detected, and the average counts were higher for the crust. The average values for water activity (a_w_) and pH were 0.98 ± 0.02 and 5.77 ± 0.015, respectively. A slight decrease in a_w_ and an increase in pH were observed over the aging period (*p* < 0.05). The lower microbial counts on the lean meat and the overall assessment of freshness confirm the importance of good manufacturing and storage practices during dry aging.

## 1. Introduction

For some time now, butchers have used the dry aging process to enhance the quality of both high- and low-grade primal beef cuts [1]. Dry aging, particularly when applied to high-grade primal cuts, results in a beef product characterized by enhanced tenderness. This has attracted a great deal of interest globally, particularly in regions such as the United States and Asia, especially in the “fine cuisine” sector. Presently, there is a growing consumer interest in dry-aged beef, creating a significant niche in the food service market [2,3]. The process involves selecting unpackaged primal beef cuts and placing them in a controlled environment room for several weeks. The main goals of dry aging meat are related to flavor intensification, resulting in beefier and nuttier flavor profiles, and to improve succulence and tenderness [4]. In addition, the dry aging process also modifies the microbiological, physical, and chemical characteristics of the beef [5]. The juices are absorbed into the meat, undergoing a chemical breakdown of protein and fat constituents [2,6]. This process involves water diffusing from the interior to the surface, followed by evaporation into the environment, ultimately leading to the concentration of flavor compounds [7].

Drying costs are higher than conventional processing methods. Firstly, there’s a notable shrinkage, trim loss, and an increased risk of contamination. Secondly, the requirements for specific aging conditions, including the need for only the highest grade of beef marbling, contributes to elevated costs [2]. The process guidelines include parameters such as the days of aging, storage temperature, relative humidity (RH) and air flow [2,7]. Typical dry aging times range from 14 to 40 days. However, this varies with temperature; lower temperatures require more time to achieve desired results [2]. The ideal temperature is between 0 °C and 4 °C, which is critical for successful dry aging. While high temperatures improve the enzymatic process, they also encourage rapid bacterial growth [2]. Controlling the process involves not only monitoring the temperature but also the RH. Elevated RH, increases the risk of bacterial spoilage, creating an unpleasantly viscous surface. On the other hand, low RH can limit bacterial growth and promote surface dehydration and weight loss [2,7]. The recommended RH range is between 61% and 85% [2].

After slaughter, during the conversion of muscle to meat, the pH drops as the hydrogen accumulates until the isoelectric point is reached. The pH of the meat may be important during the dry aging process as it affects the muscle’s ability to bind water [7]. Regarding dry-aged meat, some studies have reported a similar pH to fresh meat, while others have found a positive correlation between pH, ranging from 5.8 to 6.9, and the growth of lactic acid bacteria (LAB) in dark cut samples [8]. Throughout the drying process, a dynamic microbiome is created on the meat surface (crust), consisting particularly of various *Lactobacillus* spp. Including *L. sakei* and *L. plantarum* [9,10]. This environment limits bacterial growth and promotes the emergence of beneficial molds [2]. The continuous activity of bacteria, yeasts and molds metabolizing and producing metabolites on the meat affects its quality and safety [6,9]. Bacteria, such as LAB can contribute to meat spoilage; for example, LAB can cause greening of meat, while *Pseudomonas* spp. Metabolizes glucose, lactate and amino acid, resulting in off-odors [6,11]. Nevertheless, there are some prevalent bacteria during the dry-aging, like LAB, yeasts and molds such as *Mucor* spp., and *Penicillium* spp. [6,8]. On the other hand, foodborne pathogenic bacteria such as *Listeria monocytogenes*, enterohemorrhagic *Escherichia coli* and *Salmonella* spp. may be present and proliferate during the dry aging process, as they are active at low temperatures [11,12]. However, studies suggest a tendency for pathogen reduction over time [8,11,12]. Despite the unknown correlation between the microbiota and the quality of dry-aged meat, studies focusing on yeasts and molds on the surface of dry-aged beef, have demonstrated their ability to improve sensory traits, by releasing proteases and collagenolytic enzymes that break down myofibrils enhancing flavor [1,6,8,11].

As consumer interest grows, so does the need for studies to support food safety. In Europe, there is currently no specific regulation for dry-aged meat with specific microbiological criteria. Regulation (EC) No 2073/2005 sets the microbiological limits for general foodstuffs and it is the only one to date. Therefore, the European Food Safety Authority (EFSA) recently published a scientific opinion on the microbiological safety of dry-aged meat [8].

Tenderness, flavor and juiciness are important factors for consumers in determining the acceptability and palatability of beef [13]. In terms of appearance, dry-aged beef has been shown to be darker with a more stable color than wet-aged beef [14,15]. Studies comparing the sensory quality of dry-aged beef are inconsistent, with some reporting no effect of dry-aged beef on sensory traits [15,16], while others, mainly in Europe, Japan, and New Zealand, have shown beneficial effects of dry aging on consumer sensory quality perception [14,15,17].

Dry-aged beef occupies a niche and expensive market, requiring not only premium beef carcasses, an exclusive and controlled process, but also high trimming and weight losses. Nevertheless, the public opinion on these products is controversial, while some unquestionably enjoy them, others express food safety concerns due to the absence of specific regulation. Hence, standardization of the process is crucial to address concerns about the safety of dry-aged beef. Therefore, this study was designed to characterize the microbiological and physicochemical changes of beef during dry aging in a Portuguese company, as well as the sensory perception of the dry-aged meat.

## 2. Materials and Methods

### 2.1. Study Planification

This study was carried out in a company certified for the meat dry aging process in Portugal in order to evaluate the safety and consumer acceptance of the dry aging process. Twelve loins (*Longissimus lumborum*) from six animals with similar characteristics (sex, age and weight) were selected for the study. In order to obtain a cut surface with the specific aging time (Figure 1) the loins were divided into three pieces (about 5.0 kg each) and the maximum dry aging time was 90 days. At the end of the study, the leftover meat was discharged.

### 2.2. Dry-Aged Process Guidelines

In the facilities, the dry aging practices of the company were followed, including controlling the refrigerator room temperature, RH and air flow, and maintaining continuous UV light. The first day (day 1), starting three days after slaughter was considered as the beginning of the aging process. Throughout the study, the temperature, and RH in the dry aging refrigerated room, ranged from 4.3 to 2.1 °C and 69.0 to 55.4%, respectively. Airflow was maintained within a programmed range of 0.5 m/s to 2.0 m/s. Before each sample collection, the intramuscular pH of each piece was measured with commercial equipment (HI98163, HANNA^®^, Póvoa de Varzim, Portugal), using the FC2323 meat pH electrode and a stainless-steel piercing blade (FC099). The environmental temperature and RH were also recorded.

### 2.3. Laboratory Analysis

#### 2.3.1. Sample Collection

On days 1, 14, 21, 35, 60 and 90 of dry aging, from a cutting surface of the piece of each loin, were hygienically taken two samples (total of 24 samples/time) and transported to the laboratory at 4 °C within 1 h 30 min. The samples were taken from two different sites of the piece surface, similar to a cut beef, with approximately 400 g (with superficial and lean meat).

#### 2.3.2. Microbiological Analysis

In the laboratory, the packed samples were placed in a refrigerated room until the preparation for the different steps. From the 12 samples, 10 g and 25 g x 2 of superficial meat (crust) were aseptically taken, and the same procedure was used for 12 samples of lean meat, to microbiological analysis. To quantify total aerobic bacterial populations and pathogens, 10 g of the samples (superficial and lean meat) were added to 90 mL of 0.1% buffered peptone water and the samples were then homogenized for 60 s using a stomacher. The homogenates were then serially diluted and 1 or 0.1 mL portions of the diluted suspensions were poured-plated by incorporation or surface-plated on non-selective and selective agar plates. To quantify different groups of bacteria, the following media and conditions were used: plate count agar at 30 °C for 72 h for the total mesophilic aerobic bacteria (ISO 4833-1), and at 7 °C for 10 days for the total psychrotrophic aerobic bacteria (ISO 174410); Baird-Parker agar at 37 °C for 48 h for coagulase-positive staphylococci (ISO 6888–2); Tryptone Bile Glucuronic agar at 44 ºC for 24 h for *E. coli* (ISO 16649–2); Violet Red Bile Glucose at 37 °C for 24 h for *Enterobacteriaceae* (ISO 21528–2); Cephalothin-Sodium fusidate-Cetrimide agar at 37 °C for 24 h for *Pseudomonas* spp. (ISO 13720); Chloramphenicol Glucose agar at 25 °C for 5 days, and Sabouraud dextrose agar at 30 °C for 7 days, for yeasts and molds (ISO 21527–1); Man–Rogosa–Sharpe at 30 °C for 3 days for mesophilic lactic acid bacteria (ISO 15214); chromogenic medium agar for detection, isolation and enumeration of *L. monocytogenes* at 37 °C for 24–48 h for *Listeria* spp. and *L. monocytogenes* (ISO 11290–2/A1). Hektoen enteric agar at 37 °C for 24 h for *Salmonella* spp. (ISO 6579-1). The search of *L. monocytogenes* was performed using 25 g, in 225 mL of Fraser I and Fraser broth and then spread on a chromogenic medium agar (ISO 11290:1998-1/AFNOR Validation CHR-21/1-12/01). Detection of *Salmonella* sp. was performed using 25 g samples in 225 mL of buffered Peptone Water medium and then with selective enrichments—Rappaport-Vassiliadis broth and Muller–Kauffman Tetrathionate-Novobiocin spreads on chromogenic medium agar and Hektoen enteric agar (ISO 6579:2017). After incubation, plates with colonies were counted using a spiral grid and the number of CFU/g was calculated and log_10_ transformed.

#### 2.3.3. Physicochemical Parameters

The color of the meat was measured on the crust and on lean meat, on the total of the 24 samples each time. The samples were measured at room temperature with a chroma meter (CR410, Konica Minolta Co., Osaka, Japan) that had been calibrated using a standard white tile. CIE L* (lightness), a* (redness), b* (yellowness) values were measured at three random locations on each sample. The a_w_ was quantified in lean meat, at day 1 and 60 (total 12/time) using an aw-kryometer (Rotronic HigroLab^®^, China).

The intramuscular pH was performed as described on Section 2.2.

### 2.4. Sensory Analysis

The sensory traits evaluation occurred each time point by a trained panel (a total of 6 members per session) and was carried out under controlled light conditions in a sensory room facility. The members of the panel were previously selected and trained in accordance with ISO 8586-1 (2001) and were familiar with sensory assessment of meat. During the session, this trained panel performed the sensory evaluation of 2 pieces from each sample (loin): one entirety (including crust and lean meat) and the other trimmed (resulting in a total of 24 evaluations per session). The assessors used a structured scale ranging from 0 to 7 according to Saraiva et al. [18]. The descriptive attributes were based on the perception of color (red and brown), viscosity, odor (intensity, sweet, buttery, rancidity) and overall assessment of freshness (0 indicating minimal and 7 indicating substantial perception).

### 2.5. Statistical Analysis

Microbiological enumeration data were log_10_ transformed for analysis. Descriptive statistics, including means and correlation values, were calculated for microflora, pH, color and sensory evaluation. To evaluate differences among distinct times, non- parametric analysis Kruskal–Wallis tests were performed, followed by Bonferroni correction for multiple pairwise comparisons. A significant level of 5% was used. Microbiological counts were visualized using violin plots generated with the ggplot2 package in R version 4.3.0. Principal components analysis (PCA) was employed to separate meat and crust samples and to explore the interrelations among variables. Correlation matrices, for crust and meat samples, were also examined to assess the importance of variables in group separation. Partial least squares-discriminant analysis (PLS-DA) was conducted to further differentiate between group types, using two components for prediction. Variable importance in projection (VIP) scores were generated and plotted, using the mixOmics package, to identify influential variables in separation of groups. The quality of the PLS-DA is given by the classification error rate (based on a 5-fold cross-validation strategy).

## 3. Results and Discussion

### 3.1. Dry-Aged Process and Meat Microbiological and Physicochemical Status

In order to evaluate the surface microbiome dynamics and the trimming effect, data were analyzed separately for the superficial beef (crust) and for lean meat (meat) with significant differences between them for most variables *(p* < 0.05).

#### 3.1.1. Microbiological Counts

At the beginning of dry aging, the microbial counts are higher on crust than in meat, significantly for Enterobacteriaceae (Table 1). During time the crust counts increase, significantly, on mesophilic bacteria, LAB, Enterobacteriaceae, *Pseudomonas* spp. and molds, with higher numbers at day 90. No significant differences were observed for yeasts (*p* > 0.05). In turn, the lean meat counts decrease with aging time, specially the mesophilic and psychrotrophic bacteria, LAB, *Pseudomonas* spp. and yeasts (*p* < 0.05). Molds increased counts with time on both samples, besides no significant differences were observed in lean meat *(p* > 0.05). The lower counts compared to the initial values and over drying on lean meat were at day 21 for mesophilic and psychrotrophic bacteria, LAB, *Pseudomonas* spp. and yeasts corresponding to molds higher peak. The lower counts on lean meat on overall traits suggest beef dry-aged optimal time of 21 days with an opposite relation between the bacteria and yeasts counts and the molds counts. At day 14 when most bacteria were low, the molds counts were higher, suggesting the dynamic microbiome of the dry-aged beef [19]. The differences in counts between the group types can be related to trimming practices. Mesophilic bacteria, LAB, yeasts and *Pseudomonas* spp. had higher median values on crust and meat group (Figure 2). The pathogens count like *Listeria* spp., *Salmonella* sp. *Staphylococcus* spp., *E. coli* were all negative on enumeration for each time and for the crust and lean meat. These results are consistent with previous studies [8,11]. The dry-aged process parameters, like refrigeration temperature, HR and forced air, meat a_w_ and pH could be a positive influence for development of food borne pathogens [11].

The significant increase in bacteria and yeast counts during dry aging has been demonstrated in other studies [4,8]. Recently, Gowda et al., reported great variation on microbiological counts between loins, once they were coming from different companies with different processes and aging times [11]. The study showed, at the end of the ripening process, high numbers of psychrotrophic bacteria, *Pseudomonas* spp., LAB and *Brochothrix thermosphacta* on both tissue types [11,20]. However, another study reported that with a long ripening period there was a decrease in LAB counts and an increase in yeasts counts and observed similar numbers in both adipose tissue and lean meat [21].

Molds were detected at 19 days of aging [11], and in the present study were detected on day 14, in contrast with other studies that referred to molds being detected 3 weeks after the initial aging time [2].

#### 3.1.2. Color, pH and Water Activity

The average a_w_ on meat was 0.985 ± 0.009 on day 1 and 0.976 ± 0.013 on day 60. The a_w_ can be related to the microbiological profile of the meat and process parameters. Some studies have reported that a low a_w_ of the beef surface may reduce the ability of bacteria to cause spoilage and the growth of pathogens [16]. Nevertheless, in this study the lower microbiological counts were at day 21 suggesting the need of further studies in order to correlate the variables. Although the lack of results over time, the present showed a tendency of a_w_ to decrease with time. Some previous studies refer to inhibition of growth of foodborne pathogens at a_w_ less than 0.93 with mold tolerance [11].

The presented dry aging process revealed effects on pH and color *(p* < 0.05). Despite the fluctuation of the values during the different times, the pH increased and differed significantly from day 1 and day 21 and until day 90 (Table 2), always with values < 6.1. Previous studies have reported no influence of pH beef drying [4,21] and significant differences with the aging method [15]. It is known that meat pH is related to Maillard reactions during cooking and therefore to meat flavor. Based on this assumption, higher pH in dry-aged beef may promote flavor improvement [15,17,21].

Significant differences on surface color (CIE L*a*b*) were found on both types of samples for a* and b* *(p* < 0.05) during aging (Table 2), suggesting the time effect on instrumental color. The crust and lean meat became less luminous (L*) but no significant differences were observed over the aging time. For redness (a*) and yellowness (b*) moderate and high significant differences were observed, respectively, in both crust and lean meat. Previous studies have revealed significant color (L*, a* and b*) changes on dry-aged beef, with lower values for L* and b* and higher for a* [14,15]. Nevertheless, the instrumental color measures reflectance (wavelengths 630/580 nm) which is close to what the eye can see, and this is correlated with differences in muscle components, moisture and pigment concentration [15].

### 3.2. Sensory Analysis of Dry-Aged Meat

The sensory analysis was mainly on beef color, odor and overall assessment of freshness, where the panel visualized the whole sample, without trimming (crust + lean meat) and the trimmed sample (meat). The loins were refrigerator stored and on each collecting day, the samples were collected from each cutting surface of the piece (explained on material and methods). The panel observations were made according to the beef appearance, with a score of 0 (not fresh) until 7 (very fresh). The present results were calculated through the average of the sensory panel on each sample and time. Figure 3 represents examples of the loin pieces used in the study, and showed the overall color differences of the dry-aged beef before and after trimming and with time effect. Regarding the sensory evaluation of the color, the panel considered the whole and trimmed samples less redness and browner *(p* < 0.05) (Table 3). The apparent viscosity increases with aging and the odors, on both sample types, the intensity increases with time *(p* < 0.05) and therefore, the sweet, buttery, and rancidity odors with predominance of sweet *(p* < 0.05). Nevertheless, the values were lower on lean meat samples (Table 3). The overall meat freshness decreases with time (*p* < 0.05), and it is higher on lean meat samples. The results are in line with the expectations, once the dry-aged process leads to evaporative losses (dehydration) and muscle oxidation (browner and less red). The sweet odor can be related to the nutty flavors of the dry-aged beef and maybe a signal of the drying quality [2]. Regarding the freshness trait, on day 90 the score was 3.48 for lean meat, showing that for the panel the dry-aged beef still fresh until 90 days. The majority of beef dry-aged sensory analysis based on eating quality attributes showed that dry-aged beef had higher tenderness, flavor and overall liking when compared to wet-aged beef [4,14]. Berger et al. reported significant palatability improvement of dry-aged beef compared with wet-aged, and the positive sensory traits impact of beef aging is reported by other studies [1,4,14,20].

### 3.3. Multivariate Analysis and Correlations

The overall variables behavior was evaluated by partial least squares—discriminant analysis (PLS-DA) (Figure 4) and principal component analysis (PCA) (Figure 5) in order to separate and distinguished differences based on each group and variables were measured by scores known by variable importance in projection (VIP) (Figure 6).

The PLS-DA showed the lean meat samples on the positive side of comp 1, more associated to positive color attributes (Figure 5), namely red color and fresh meat. It also showed the crust samples (points) more related to comp 2 on its positive side, associated to the microbiota of spoilage, especially mesophilic and psychrotrophic bacteria and *Pseudomonas* spp. with significant importance (Figure 5).

Figure 5 illustrates the biplot for principal components: the samples were separated by type (crust or meat) along PC1 and aging deteriorative process (PC2). The relations between the variables are shown on PC1:PC2 and PC2:PC3, and together represented 79.6% of the explained variance. On crust samples, the microbiological traits are strongly related and located in the positive side of PC2, increased with drying. On other hand, the color L* contributes strongly to PC3 and is associated with crust samples. On meat samples the mainly variables related are sensory positive attributes red color and freshness in the positive side of PC1, associated to the fresh samples (days 1, 14 and 21), positively correlated with the CIE a*, and negatively associated with the undesirable attributes (brown color, off odors intensity) on the opposite side. The color a* has a high relation with group meat and the values decrease with aging.

Figure 6 shows variables with significant importance: LAB, *Pseudomonas* spp. yeasts and sensory attributes namely sweet, buttery, and rancid odors and brown color on crust and CIE a*, b* and red color attribute on lean meat. These results could mean that the color of the meat (CIE a*—redness) is highly related to aging, decreasing with time (Table 2). On crust, the LAB and yeasts could be a microbiological indicator to aging, increasing with time showing significant differences, except for yeasts on crust samples as observed by Bischof et al. [22].

In addition to PCA, partial least squares-discriminant analysis (PLS-DA) was performed to separate between groups of observations, and it can discriminate differences based on each group. In this study, PLS-DA showed a low classification error rate of 5.9%, when we used train data, only six samples were not well classified, one from the crust and five from lean meat. With the test validation sample the result is better with only one meat sample (2.4%) not well classified. These results show a good predictive ability as observed in other studies [23]. The aging period were clearly distinguished based on the color variables, specially, a* and red color more related with freshness in the lean meat; and the microbiological variables and off-odors in the crust.

## 4. Conclusions

The current process shows that aging loins at <4.0 °C with RH < 65.0% can be safe from a microbiological perspective. This study reveals high crust counts with aging on mesophils, psychrotrophics, LAB, *Pseudomonas* spp. with average values of 4.7 and 5.8 Log_10_CFU, respectively. On lean meat the counts were, in general 2–3 Log lower than in crust. Molds only present growth and increase on crust samples during time, similar to the references.

The meat color (CIE a*—redness) is highly related with aging, decreasing with time with the crust becoming browner. The sweet odor was the most predominant in the end of aging in both sample types. Rancid and buttery odors are higher in crust samples than in lean meat, showing significant differences. This can be related to the meat freshness score of 3.48 for lean meat, indicating that for the panel the dry-aged beef was still fresh until 90 days. Nevertheless, the wide range of parameters that is used in practice and the higher loads that were occasionally found, suggests that future studies should also assess microbiome communities and their influence on the growth/survival of pathogens. Therefore, we cannot completely rule out the potential growth of psychrotrophic pathogens during storage, though this still needs to be assessed. Hygiene during retail practices and proper storage of the end product remains of critical importance and need further studies to assess the quality and safety of the end product.

The present data reveal that drying did not adversely impact the surface meat color and overall assessment of freshness immediately after the trimming. Further research in characterizing more descriptive sensory attributes of the dry-aged beef loin by a trained/certified sensory panel, conducting lipid/protein oxidation measurements and/or identifying volatile compounds governing the unique flavor of dry-aged beef loins would be warranted.

## Figures and Tables

**Figure 1 foods-12-04514-f001:**
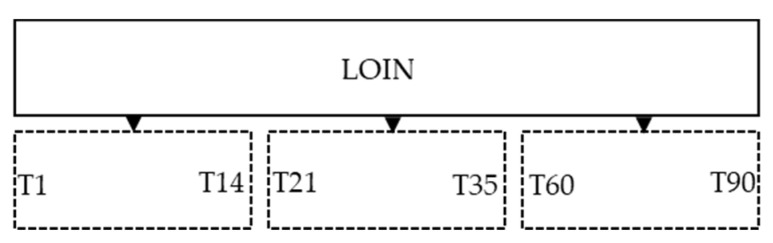
Scheme of the study design of the cutting surface during the aging plan.

**Figure 2 foods-12-04514-f002:**
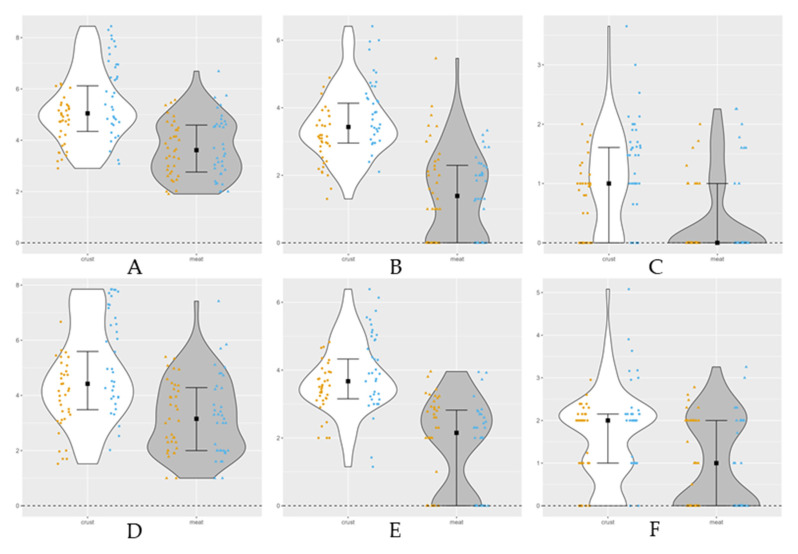
Violin plot of dry-aged beef microbiological counts on crust and lean meat with a detection limit of 1 log_10_ CFU/g (lower dashed line). The separated samples, crust (*n* = 72) and meat (*n* = 72) are presented by mirrored density plots including the median and first and third quartile. Points (crust) and triangles (meat) represented individual observations at dry aging beginning (yellow) and ending (blue). (**A**) Total mesophilic bacteria; (**B**) lactic acid bacteria; (**C**) *Enterobacteriaceae*; (**D**) molds; (**E**) yeasts; (**F**) *Pseudomonas* spp.

**Figure 3 foods-12-04514-f003:**
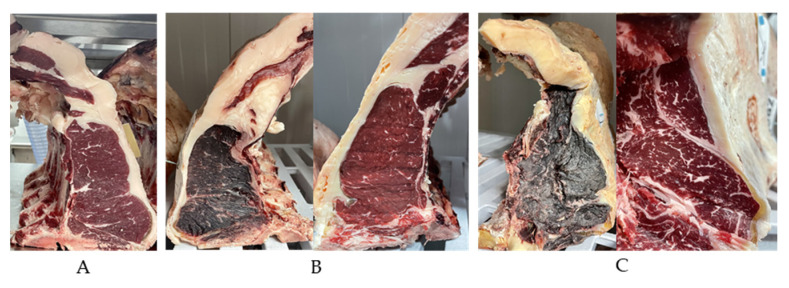
Beef loin used for the study. (**A**) Cutting surface at day 1; (**B**) cutting surface at day 35 (left image represents the crust and right image represents the meat after trimming); (**C**) cutting surface at day 60 (left image represents the crust and right image represents the meat after trimming).

**Figure 4 foods-12-04514-f004:**
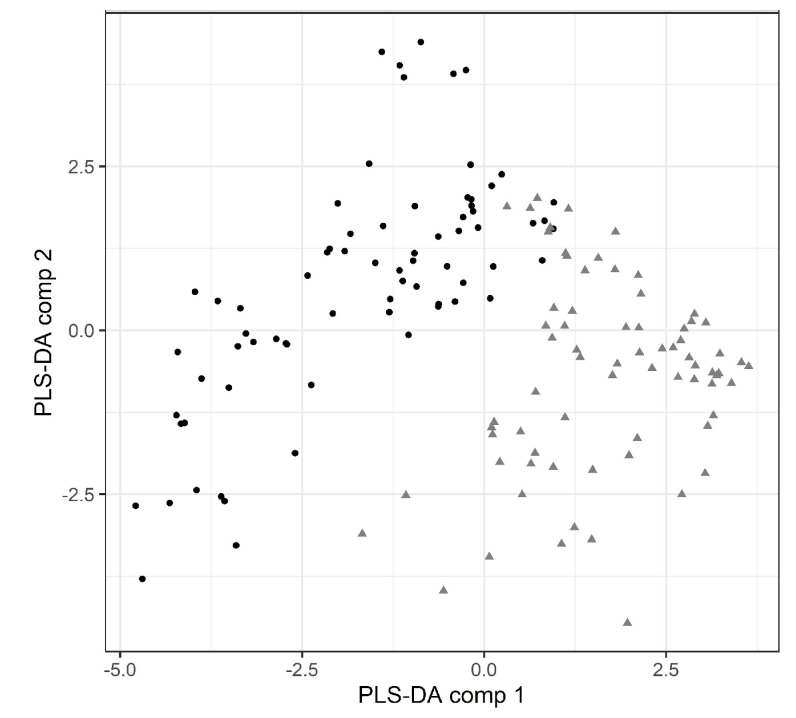
Partial least squares-discriminant analysis (PLS-DA) from the crust samples (points) and lean meat samples (triangles).

**Figure 5 foods-12-04514-f005:**
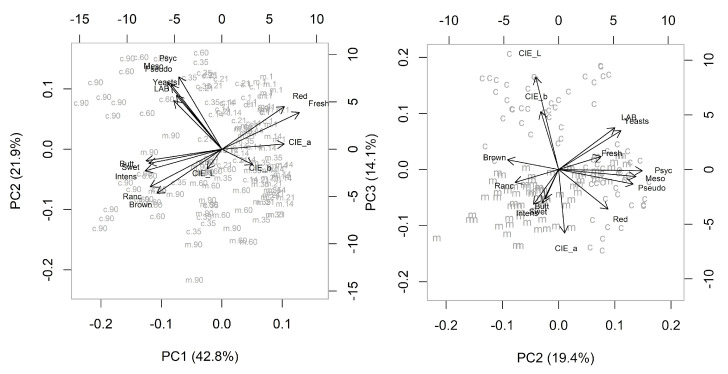
Principal components analysis (PCA) biplots illustrating the sample type variation, crust (c) and meat (m) (PC1) and the aging time (PC2; PC3) on variable patterns: mesophilic, psychrotrophic and LAB, *Pseudomonas* spp., yeasts, CIE L*, a*, b*, sensory analysis: redness, brown color, sweet, buttery, rancid odors and its intensity, and freshness.

**Figure 6 foods-12-04514-f006:**
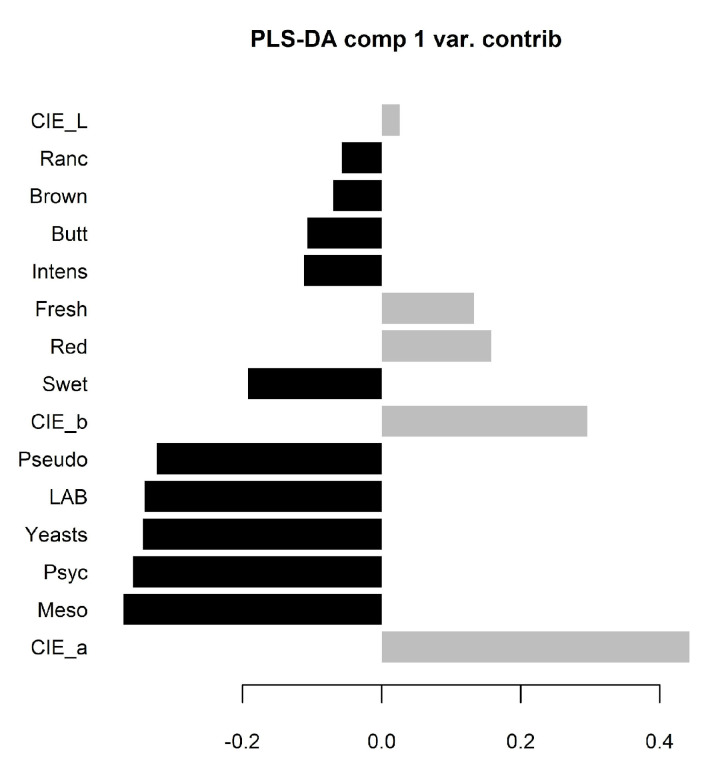
Partial least squares—discriminant analysis (PLS-DA) demonstrating variable importance projection (VIP) scores from the crust (black columns) and meat (grey columns) from the train data 5-fold cross-validation.

**Table 1 foods-12-04514-t001:** Microbiological counts (means), expressed in Log_10_ CFU/g, on crust and lean meat, accordingly aging time and sampling type (crust and lean meat).

Trait	Type	1 d		14 d		21 d		35 d		60 d		90 d		*p*
Mesophiles	C	4.940	^ab^	4.310	^b^	4.870	^ab^	5.100	^ab^	6.150	^ab^	6.240	^a^	*
	M	4.600	^a^	3.330	^bc^	2.980	^c^	3.400	^abc^	3.550	^abc^	4.340	^ab^	**
	*P*	NS		*		***		**		***		*		
Psycrothrophics	C	4.740	^a^	4.540	^a^	5.300	^a^	4.780	^a^	5.880	^a^	5.720	^a^	NS
	M	4.500	^a^	3.310	^a^	3.250	^a^	3.410	^a^	3.330	^a^	3.550	^a^	*
	*p*	NS		*		***		NS		**		***		
Lactic Acid Bacteria	C	3.600	^ab^	2.460	^c^	3.190	^bc^	3.500	^ab^	3.690	^ab^	4.740	^a^	***
	M	3.150	^a^	0.440	^c^	1.050	^bc^	1.260	^bc^	1.680	^b^	1.240	^bc^	***
	*p*	NS		***		***		***		***		***		
Enterobacteriaceae	C	0.840	^abc^	0.670	^bc^	0.490	^c^	1.320	^ab^	1.360	^ab^	1.710	^a^	***
	M	0.280	^a^	0.390	^a^	0.380	^a^	0.730	^a^	0.380	^a^	0.150	^a^	NS
	*p*	*		NS		NS		NS		**		***		
*Pseudomonas* spp.	C	3.800	^ab^	3.410	^b^	4.670	^ab^	4.710	^ab^	5.600	^a^	5.820	^a^	**
	M	4.290	^a^	2.750	^ab^	2.640	^b^	2.940	^ab^	3.190	^ab^	3.640	^ab^	*
	*p*	NS		NS		**		**		**		*		
Yeasts	C	3.500	^a^	3.270	^a^	3.620	^a^	3.670	^a^	4.320	^a^	4.160	^a^	NS
	M	3.120	^a^	1.720	^b^	1.260	^b^	1.610	^b^	1.330	^b^	2.020	^ab^	*
	*p*	NS		**		***		***		***		**		
Molds	C	0.830	^b^	1.920	^a^	1.350	^ab^	1.520	^ab^	2.040	^a^	2.320	^a^	**
	M	0.690	^a^	1.170	^a^	1.510	^a^	0.690	^a^	0.940	^a^	1.020	^a^	NS
	*p*	NS		*		NS		NS		*		*		

In each row, means with different letters differ significantly *(p* < 0.05). * *p* < 0.05; ** *p* < 0.01; *** *p* < 0.001; NS: non-significant. C—crust sample; M—lean meat.

**Table 2 foods-12-04514-t002:** Means of instrumental surface color and pH on crust and lean meat, accordingly aging time and sampling type (crust and lean meat).

Variable	Type	1 d		14 d		21 d		35 d		60 d		90 d		*p*
pH	C	------		------		------		------		------		------		
	M	5.610	^c^	5.770	^bc^	5.800	^ab^	5.850	^ab^	5.740	^ab^	6.010	^a^	***
CIE L*	C	57.520	^a^	46.590	^a^	51.660	^a^	48.660	^a^	52.400	^a^	45.950	^a^	NS
	M	35.110	^a^	36.060	^a^	36.370	^a^	35.410	^a^	34.340	^a^	34.110	^a^	NS
	*p*	***		**		*		*		NS		NS		
CIE a*	C	13.350	^a^	10.720	^ab^	11.520	^a^	8.250	^ab^	6.460	^ab^	3.770	^c^	**
	M	24.460	^ab^	24.440	^a^	24.380	^ab^	22.740	^ab^	21.820	^b^	16.560	^c^	***
	*p*	***		***		***		***		***		***		
CIE b *	C	14.740	^a^	10.460	^bc^	12.650	^ab^	15.540	^ab^	12.920	^ab^	7.480	^c^	***
	M	14.170	^a^	13.800	^ab^	12.920	^abc^	12.580	^bc^	12.080	^cd^	10.550	^d^	***
	*p*	NS		**		NS		NS		NS		***		

In each row, means with different letters differ significantly *(p* < 0.05). * *p* < 0.05; ***p* < 0.01; ****p* < 0.001; NS: non-significant. C—crust sample; M—lean meat.

**Table 3 foods-12-04514-t003:** Sensory attributes of the whole beef (C + M) and meat (M) by the trained panel.

Sensory Attributes		Type	1 d		14 d		21 d		35 d		60 d		90 d		*p*
Color	Redness	C + M	5.810	^a^	5.100	^ab^	4.860	^b^	3.770	^bc^	2.920	^c^	2.690	^c^	***
		M	5.990	^a^	5.620	^ab^	5.420	^bc^	4.690	^cd^	4.480	^d^	4.660	^d^	***
		*p*	NS		**		**		NS		NS		**		
	Brown	C + M	0.080	^c^	0.910	^bc^	1.640	^b^	3.710	^a^	4.450	^a^	4.400	^a^	***
		M	0.230	^d^	1.050	^cd^	1.760	^bc^	2.330	^ab^	3.060	^a^	3.300	^a^	***
		*p*	NS		NS		NS		*		*		NS		
Viscosity		C + M	0.200	^c^	0.530	^abc^	0.340	^abc^	0.280	^bc^	0.940	^ab^	1.670	^a^	**
		M	0.250	^b^	0.670	^b^	0.610	^b^	0.650	^b^	1.260	^a^	1.460	^a^	***
		*p*	NS		NS		NS		**		NS		NS		
Odor	Intensity	C + M	0.450	^d^	1.180	^c^	1.550	^c^	2.300	^b^	3.340	^a^	4.380	^a^	***
		M	0.220	^d^	0.910	^c^	1.340	^bc^	0.600	^b^	2.940	^a^	3.250	^a^	***
		*p*	NS		NS		NS		*		NS		*		
	Sweet	C + M	0.800	^c^	1.180	^c^	1.260	^c^	1.760	^b^	2.080	^b^	3.580	^a^	***
		M	0.840	^c^	0.820	^c^	1.160	^bc^	0.550	^bc^	1.660	^b^	3.120	^a^	***
		*p*	NS		**		NS		*		NS		NS		
	Buttery	C + M	0.610	^c^	1.140	^bc^	1.300	^b^	1.250	^b^	2.870	^a^	2.990	^a^	***
		M	0.680	^c^	0.970	^bc^	1.180	^b^	0.390	^b^	1.840	^a^	2.200	^a^	***
		*p*	NS		NS		NS		NS		**		*		
	Rancid	C + M	0.110	^c^	0.570	^b^	0.680	^b^	0.950	^b^	1.360	^b^	3.110	^a^	***
		M	0.150	^d^	0.480	^c^	0.620	^bc^	0.410	^c^	1.200	^ab^	2.340	^a^	***
		*p*	NS		NS		NS		NS		NS		*		
Freshness		C + M	6.320	^a^	5.240	^b^	5.270	^b^	4.300	^c^	3.700	^c^	2.580	^d^	***
		M	6.380	^a^	5.640	^b^	5.540	^b^	0.360	^c^	4.280	^d^	3.480	^d^	***
		*p*	NS		**		NS		**		NS		*		

In each row, means with different letters differs significantly *(p* < 0.05). * *p* < 0.05; ***p* < 0.01; ****p* < 0.001; NS: non-significant. C—crust sample; M—lean meat.

## Data Availability

Data are contained within the article.

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
