# Peer review of "Microbial, Physicochemical Profile and Sensory Perception of Dry-Aged Beef Quality: A Preliminary Portuguese Contribution to the Validation of the Dry Aging Process"

_foods, 2023, doi:10.3390/foods12244514_

Round 1

Reviewer 1 Report

Comments and Suggestions for Authors

This study conducted to explore the changes of microbiological and physicochemical traits of dry meat during 90 days of aging. This trail is of great significance to promote the technique of dry aging process. However, the language needs to be improved and more discussion was needed to deepen the novelty of this study. The major concern is number of the consumer panel which is less to represent the Portuguese consumers. Some other comments are listed below:

Line30-31 This seems to explanation for the microbiological count and it is suggested to be deleted in the abstract.

Line 33-34 it is acknowledged that importance of good trimming and storage practice for the dry aged beef, however, this is not the aim of this study and no treatment for trimming is conducted.

Line 45-47 this is a long sentence while no conjunction is used.

Line 52 RH indicates relative humidity ?

Line 64 what are others ?

Line 115 specify the details of those parameters

Line 166 when using untrained consumer panel, a total of 6 are nothing much and it is not convincing to do the sensory analysis.

Line 174-176 when comparing the difference between aging times, it is importance to regard sample location of loin as a random effect.

Line 188-189 those should be included in section 2.2

Line 244 what is the reason for no detection of aw during aging ?

Line 310 Please indicate the variables that were used for PLS-DA and PCA analysis? In figure 5, pH value was not found.

Line 324 Three components explained a total of 63.1%% of variance indicating the PCA model is not good.  

Comments on the Quality of English Language

The language needs to be improved

Author Response

Dear reviewer,

The authors thanks the valuable comments and suggestions of the Reviewer that allowed the improvement of manuscript quality. All the revisions are marked in the text.

Best regards,

Cristina Saraiva

Reviewer 2 Report

Comments and Suggestions for Authors

After reading the manuscript "Microbial, physicochemical profile and sensory perception of dry aged beef quality and safety: a Portuguese preliminary contribu- tion for the dry ageing process validation", I realized that the manuscript showed in some parts the scientific rigour wanted, but in other parts I have missed it.

The authors have presented critical evaluation only in some paragraphs.

The references are not exactly current, besides Introduction, rationale, methods should be improved.

Thats why I have written some suggestions below in an attempt to improve the paper.

L.3- What would you have evaluated as "safety"? Is it not covered by the microbiology section? Re-evaluate the need for this word.

L.4- What would you have thought of "process validation"? 

I know validation as another way of approaching projects, when you want to evaluate an instrument in different groups, different cultures... I don't think that is the case here.

Review the need for these terms in the title.

L.23-Prefer sensory instead of "sensorial"

L..35- Avoid using "sensorial"

L.40- I missed mentioning some countries that consume dry aged beef, as well as how it is consumed. I also suggest highlighting the nutritional importance and price of this product.

L.52- Has this acronym been defined previously?

L.67- Please, be careful with scientific names in italics

L.74- since they are pathogenic, they could offer risks and they weren't mentioned, right?

L.78 - "studies show positive effects" - Which effects ?

L.106- L. cientific name 

L.107- were instead of where

L.123- Why weren't the periods equidistant?

L.164- I missed the texture analysis, I didn't find it either in the physical nor in the sensory analysis

L. 165- For a sensory test a lot of relevant information was not included, I suggest reading  other papers and improving your article. 

Has the project been submitted to an evaluation by a university ethics committee? Did it follow the Helsinki declaration? Please, enter the approval protocol number.   Which sensory test was performed? How many assessors ? What was the profile of the assessors ? How many men/women ? age of the assessors ? Are the assessors usually consumers of this product  ?   Were the analyses performed in sensory booths ? Did the assessors receive water to rinse the taste buds ?  "The sensory analysis scope was to evaluate the per- 168 ception of the raw meat, as color, smell, intensity and overall acceptability with a classification score of 0-7 (0-less; 7-lot)" - Author for this scale ? Authors, the sensory part needs to be improved, a lot of important things are missing. It can't remain like this.  

L.231-  I confess that I miss a comparison of these results with the legislation of your country, especially when you mention "safety" in the title.  

L.232-  What about day zero ? Would  it have been important? Shouldn't it be in the study limitations?  

L.233- It seems to me that you could insert in the footnote what the acronyms C ;M   stand for.   

L.270- I honestly don't know if I would include P in the "types", it is polluting the tables more than helping comprehending the results, if they are very important to you, insert them in the text in parenthesis;  

L.306- Not all the attributes that are here are in the material and methods, please review them.  

L.343- Authors, you could improve figures 4 and 5 a lot more. They're very difficult to read. I'm sorry, but if you don't follow the text, it's almost impossible to understand. It seems to me that you also need to bring into the discussion why PC1 and PC2; PC2 and PC3 were low.

L.378- The sensory part in the conclusion was very much less than expected, mainly due to the various analyses presented in the paper.

Comments on the Quality of English Language

Moderate editing of English language required

Author Response

Dear Reviewer,

The authors are grateful to the referee for the attentive and detailed remarks which helped to considerably improve the paper.

We hope the answers below and modifications introduced in the manuscript are clear and concise enough as required by the Reviewer.

All the revisions are marked in the text.

Best regards,

Cristina Saraiva

Round 2

Reviewer 1 Report

Comments and Suggestions for Authors

The authors have made sufficiently corrections in line with my comments. Thanks for that and I think the manuscript has been much improved. However, the authors have changed the untrained consumer panel to the trained panel, the training and detailed information of panels including sex, age, education level, and etc should be provided. Some points on consumer panel such as Line 109, Line 295-296, Line395, Line 405-406 needs to be rephrased.

Author Response

DETAILED RESPONSE TO REVIEWER 1

Comments and Suggestions for Authors

The authors have made sufficiently corrections in line with my comments. Thanks for that and I think the manuscript has been much improved.

Our reply: The authors are grateful to the reviewer for the valuable comments and suggestions that allowed the improvement of manuscript quality. All the revisions are marked in the text in blue color.

Reviewer´s comment: However, the authors have changed the untrained consumer panel to the trained panel, the training and detailed information of panels including sex, age, education level, and etc should be provided.

Our reply: The detail information was addicted in the manuscript.

Reviewer´s comment: Some points on consumer panel such as Line 109, Line 295-296, Line395, Line 405-406 needs to be rephrased.

Our reply: Amended as required.

Reviewer 2 Report

Comments and Suggestions for Authors Dear authors,    After another evaluation of the manuscript, I srealized a great improvement in the quality of the paper. The authors have accepted almost all of my requests.  They improved  methodology and corrected tables and graphs.  English is always useful to ask a native speaker for a final appreciation.    I would like to emphasize that I read the suggestions of other colleagues who also reviewed the paper and the corrections made by the authors and this final paper version is definitely much better.     Some notes:   L.164- I missed the texture analysis, I didn't find it either in the physical nor in the sensory analysis Our reply: Unfortunately, we didn´t done it. Answer: include in the Study limitations.   The project was not submitted to an evaluation by a university ethics committee. The assessors were trained in this type of sensory analysis and participate in other studies with raw beef meat. Answer: In my country, any paper with humans has to be submitted and approved by the ethics committee, even patients data in medical records. Please, check if you are really covered. It's the researcher's safety too.   L.186 new version - "The descriptive attributes" - you didn't mention which test was carried out, only that the attributes were descriptive. More information could be added in the Material and methods about the sensory part.   L.231- I confess that I miss a comparison of these results with the legislation of your country, especially when you mention "safety" in the title. Our reply: Totally Agree. the title was reformulated.   Answer: I did not find Legistation approcah in the new version as well. Readers will miss it for sure. Comments on the Quality of English Language

Minor editing of English language required

Author Response

Dear Reviewer,

DETAILED RESPONSE TO REVIEWER 2

Comments and Suggestions for Authors

Reviewer´s comment:

Dear authors, after another evaluation of the manuscript, I realized a great improvement in the quality of the paper. The authors have accepted almost all of my requests.  They improved  methodology and corrected tables and graphs. 

English is always useful to ask a native speaker for a final appreciation.   

I would like to emphasize that I read the suggestions of other colleagues who also reviewed the paper and the corrections made by the authors and this final paper version is definitely much better.    

Our reply: The authors are grateful to the reviewer for the valuable comments and suggestions that allowed the improvement of manuscript quality. All the revisions are marked in the text in blue color.

Reviewer´s comment: L.164- I missed the texture analysis, I didn't find it either in the physical nor in the sensory analysis Our reply: Unfortunately, we didn´t done it. Answer: include in the Study limitations.  

Our reply: The study limitation was introduced on line 321-326.

Reviewer´s comment: The project was not submitted to an evaluation by a university ethics committee. The assessors were trained in this type of sensory analysis and participate in other studies with raw beef meat. Answer: In my country, any paper with humans has to be submitted and approved by the ethics committee, even patients data in medical records. Please, check if you are really covered. It's the researcher's safety too.  

Our reply: You are totally right. However, the sensory analysis was conducted in a laboratory approved by the ethics committee (with the same trained group).

This indication was introduced as follow:

Institutional Review Board Statement
The study was conducted in accordance with the Declaration of Helsinki, and approved by the Institutional Ethics Committee) of Universidade de Trás-os-Montes e Alto Douro (Doc 99-CE-UTAD-2021).
Informed Consent Statement
Informed consent was obtained from all subjects involved in the study.

Reviewer´s comment: L.186 new version - "The descriptive attributes" - you didn't mention which test was carried out, only that the attributes were descriptive. More information could be added in the Material and methods about the sensory part.  

Our reply: The descriptive attributes were visual perception of color and viscosity; the odor and the overall acceptability was based on these physical senses.

Reviewer´s comment: L.231- I confess that I miss a comparison of these results with the legislation of your country, especially when you mention "safety" in the title. Our reply: Totally Agree. the title was reformulated.   Answer: I did not find Legistation approcah in the new version as well. Readers will miss it for sure.

Our reply: In Europe, until now, there is no specific legislation for dry aged meat. This year, EFSA emitted the first scientific opinion of the subject. This fact was included on the manuscript (line 234).